# A Novel RNA Virus in the Parasitoid Wasp *Lysiphlebus fabarum*: Genomic Structure, Prevalence, and Transmission

**DOI:** 10.3390/v12010059

**Published:** 2020-01-03

**Authors:** Martina N. Lüthi, Christoph Vorburger, Alice B. Dennis

**Affiliations:** 1Institute of Integrative Biology, ETH Zürich, Universitätstrasse 16, 8092 Zürich, Switzerland; christoph.vorburger@eawag.ch (C.V.); alicebdennis@gmail.com (A.B.D.); 2Eawag, Swiss Federal Institute of Aquatic Science and Technology, Überlandstrasse 133, 8600 Dübendorf, Switzerland

**Keywords:** RNA virus, viral transmission, *Lysiphlebus fabarum*, aphid parasitoid

## Abstract

We report on a novel RNA virus infecting the wasp *Lysiphlebus fabarum*, a parasitoid of aphids. This virus, tentatively named “*Lysiphlebus fabarum* virus” (LysV), was discovered in transcriptome sequences of wasps from an experimental evolution study in which the parasitoids were allowed to adapt to aphid hosts (*Aphis fabae*) with or without resistance-conferring endosymbionts. Based on phylogenetic analyses of the viral RNA-dependent RNA polymerase (RdRp), LysV belongs to the Iflaviridae family in the order of the Picornavirales, with the closest known relatives all being parasitoid wasp-infecting viruses. We developed an endpoint PCR and a more sensitive qPCR assay to screen for LysV in field samples and laboratory lines. These screens verified the occurrence of LysV in wild parasitoids and identified the likely wild-source population for lab infections in Western Switzerland. Three viral haplotypes could be distinguished in wild populations, of which two were found in the laboratory. Both vertical and horizontal transmission of LysV were demonstrated experimentally, and repeated sampling of laboratory populations suggests that the virus can form persistent infections without obvious symptoms in infected wasps.

## 1. Introduction

The rise of next-generation sequencing technologies has enabled *de novo* sequencing of wild and non-model organisms, which has greatly facilitated virus discovery [1]. With this has come a mounting number of studies reporting novel viruses infecting all taxa, including insects [2,3,4,5,6]. Viruses infecting insects are of great applied importance because insects can act as vectors of viral diseases affecting humans, animals, and plants [7,8] because many beneficial insects, such as honeybees, are affected by viral infections [9,10,11,12], and because insect viruses can be implemented for the control of pest species [13,14]. Viral infections also have the potential to greatly impact an important group of insects with a very specialized lifestyle, the parasitoids.

Across all insects, a large number of species employ a parasitoid lifestyle, with up to 20% of all insects belonging to the parasitoid wasps [15,16]. As adults these species live freely, but their offspring develop on or inside a host and usually kill the host during the course of their development [17,18]. *Lysiphlebus fabarum* (Hymenoptera: Braconidae: Aphidiinae) is a parasitoid of aphids, in particular the black bean aphid, *Aphis fabae* (Hemiptera: Aphididae). Female wasps inject eggs into the aphid’s body [19], where the parasitoid larva develops and finally kills the host, spinning a cocoon inside the host body and only leaving the aphid exoskeleton “mummy” when it emerges fully developed after metamorphosis. Both sexually (arrhenotokous) and asexually (thelytokous) reproducing lines of *L. fabarum* exist [20]. In the field *L. fabarum* is the most important parasitoid of *A. fabae* [21]. It imposes strong selection on *A. fabae* and contributes to the natural control of this pest aphid [22]. Interestingly, aphids are often protected against natural enemies by maternally inherited, facultative bacterial endosymbionts [23]. The most prevalent endosymbiont is the bacterium *Hamiltonella defensa,* which increases the resistance of aphids, including *A. fabae*, against parasitoid wasps [24,25]. Protection against parasitoids by *H. defensa* has been linked to the presence of a toxin-encoding bacteriophage called APSE in the *H. defensa* genome [26,27]. Viral presence in parasitoids attacking these aphids has been little explored.

While viruses are typically viewed solely as pathogens, their interactions with hosts can range from parasitism to commensalism, and even mutualism in insects [3,28,29,30,31]. Polydnaviruses (PDVs) were one of the first viral entities shown to be beneficial for their parasitoid carriers [32]. These double-stranded DNA viruses occur in some parasitoid wasps and are injected with the egg during oviposition to aid in the suppression of the host insects’ immune defense and produce optimal conditions for offspring development [33]. A separate group of viruses that has been shown to infect parasitoids are the RNA viruses [34]. The genetic material of this emerging group is encoded only in RNA-form; they lack any DNA intermediate phase during their life cycle. The possible impacts of these viruses are only beginning to be investigated in insects [34]. They either seem to have little impact on their wasp hosts, as has been reported for RNA viruses detected in parasitoids of the genus *Nasonia* [30] and *Venturia* [29], or they may aid parasitoid infection success, as recently detected in the ladybug parasitoid *Dinocampus coccinellae* [3]. The presence of the *Dinocampus coccinellae* paralysis virus (DcPV) in both wasp venom and in the neural tissue of infected ladybugs is associated with host paralysis behavior [3]. By inducing this behavior, the wasp can better ensure the development and emergence of its offspring [3].

Here we present evidence for a previously unknown RNA virus, the *Lysiphlebus fabarum* virus (LysV), isolated from the parasitoid wasp *L. fabarum* and bearing sequence similarities to DcPV. This virus was first identified in transcriptomic sequences obtained from experimentally evolved laboratory populations of the parasitoid *L. fabarum* [35]. These populations were reared on aphid hosts that either did or did not harbor different strains of the protective endosymbiont *H. defensa*, resulting in wasp populations with improved abilities to infect *H. defensa*–protected hosts [35,36]. Following this experimental evolution, putative viral proteins were found to be more highly expressed in some experimentally evolved parasitoid lines. Here, we have developed a qPCR assay for viral identification and quantification, assembled the complete viral polyprotein from RNA-seq data, and established its phylogenetic relationship to other RNA viruses. We also confirm the occurrence of this virus in wild populations, document its persistence through time in the laboratory, and experimentally demonstrate both horizontal and vertical viral transmission.

## 2. Materials and Methods

### 2.1. Insect Material

Wasps from several sources were screened for viral presence, sampled both directly from the field and from lab-reared populations that originated in the wild. The virus was first discovered in RNA-seq data generated in conjunction with an experimental evolution study designed to investigate parasitoid adaptation to the presence of the defensive endosymbiont *H. defensa* in aphid hosts [35]. Briefly, four replicate sexual populations of *L. fabarum* were reared on each of three different types of hosts: (1) a *H. defensa*-free clone of *A. fabae* (clone ID A06-407, hereafter H-), (2) a subline of the same aphid clone carrying a heritable infection with *H. defensa* strain H76 (clone ID A06-407^H76^, hereafter H76), and (3) a subline carrying *H. defensa* strain H402 (clone ID A06-407^H402^, hereafter H402). The study ran for 21 generations of experimental evolution, after which the four replicates of each treatment were merged into single populations of each type that continued to be reared on their respective hosts. We sampled wasps from these merged populations on two occasions, in April 2016 (41 generations from the start of experimental evolution), and in October 2016 (57 generations). To disentangle a potential impact of *H. defensa* on viral reproduction, the two lines reared on *H. defensa*-infected aphids (H76 and H402) were also sampled following one generation of rearing in *H. defensa*-free aphids (H-). Sampled parasitoids were reared by placing five wasps of mixed sex on plants containing week-old aphids and collecting their offspring (Appendix A). We tested for viral presence in aphids by screening individuals that had not been exposed to wasps. All three aphid lineages were sampled (H-, H76, and H402) in three replicate samples, each containing five aphids.

We also screened for viral presence in laboratory stocks of *L. fabarum*, which can reproduce both sexually and asexually [37]. Five asexual isofemale lines were screened, all collected from *A. fabae* in Switzerland between 2006 and 2009 (IL06–242; IL07–64; IL06–680; IL09–402; IL09–554). We also screened the large, outbred sexual population that was used as the base population to found the experimental evolution lines in the study of Dennis et al. [35]. This population was started from a mixture of nine stocks collected in 2012 from six locales across Switzerland, which were maintained separately for 24–30 generations before merging [35]. The population has since been maintained at a large effective population size by transferring 500 individuals every generation. In all cases, wasps were collected live from the laboratory stocks and directly frozen at −80 °C.

Lastly, to verify whether this newly detected virus also occurs in the wild, we screened individual *L. fabarum* s. str. collected across Europe (Czech Republic, England, France, Germany, Italy, and Switzerland) during four separate sampling campaigns (Appendix A). These samples were collected in 2006 [20], 2009 [36,38], 2012 [35] and 2016 [39]. Importantly, the 2012 samples included the field samples used to found the experimental evolution populations. From these 2012 samples we screened wasps that were preserved immediately after field collection, as well as wasps from these same lines after 24–30 generations of lab maintenance, but before mixing them to form the base population for experimental evolution (Appendix A). This allowed us to pinpoint whether the virus was brought into the lab through a wild population or was previously present in the lab and infected the wasp populations after their introduction to the lab. All wild-collected wasps were obtained by sampling mummified aphids and collecting wasps upon their emergence in the lab. Samples were preserved in ethanol at –20 °C, except for the 2016 collections [39], which were frozen fresh at −80 °C.

To verify that our viral assays could detect any RNA after long-term storage of wasp samples, we sequenced two known wasp genes from cDNA generated from both fresh and stored samples. For this we chose one gene that was expected to be highly expressed, the mitochondrial *CO1* gene [40], and one *L. fabarum* specific microsatellite that we did not expect to be transcribed, Lysi02 [41]. For these tests, extracted RNA was used to generate cDNA both with and without a DNase treatment, and was amplified to test for false positives resulting from residual DNA in the extraction. Thus, amplification of Lysi02 and *CO1* in samples without DNase treatment would point to the presence of DNA residues in the RNA extractions. Conversely, *CO1* should be amplifiable in reverse transcribed and DNase treated samples as this excludes any DNA residues from being amplified, and would indicate that preserved samples contain suitable RNA for testing.

### 2.2. PCR-Based Viral Screening

RNA extractions of wasps was performed on either single whole body wasps or separate ovary/body dissections according to the manufacturer’s instructions, by grinding the sample in 500 µL of Trizol reagent (Invitrogen, Carlsbad, CA, USA, catalog #15596). First-strand cDNA was synthesized according to the manufacturer’s protocol (GoScript reverse transcription system, Promega Corporation, Madison, WI, USA, catalog #A5000). Viral presence/absence was determined using PCR, carried out with the first strand cDNA products using the GoTaq Green Master Mix (Promega Corporation, Madison, WI, USA, catalog #M7122) in 10 µL reactions and the following newly designed viral primers (5′ - 3′): LysV forward ATTTTCAGAGCTCCGTGGCA, LysV reverse CTGAAGCCCGAACAAAAACG. The thermal regime consisted of 15 min at 95 °C, with 27 subsequent cycles of the following three steps: 30 s at 94 °C, 1.5 min at 58 °C, 1 min at 72 °C followed by a final 30 min extension at 60 °C. PCR products were viewed on a 1.4% agarose gel stained with PeqGreen (VWR Lab Services GmbH, Radnor, PA, USA, CAS #37–5010) and Sanger sequenced (GATC Biotech AG, European Custom Sequencing Centre, Köln, Germany) to verify the target sequences.

### 2.3. Real-Time Quantitative PCR

A real–time quantitative PCR (qPCR) assay using a Taqman fluorescent probe was developed to quantify viral RNA in a subset of the individual wasps tested with our presence/absence PCR screen (above). All reactions were scaled to a reference gene: glyceraldehyde–3–phosphate dehydrogenase (*GAPDH*), an enzyme involved in glycolysis and expected to be continuously transcribed. *GAPDH* showed no differential expression across the wasp lines and treatments used here [35], and its stable expression is supported by previous studies with virally- or bacterially-infected insects [42,43,44,45]. The aforementioned viral primers (LysV forward, LysV reverse) were used to target the virus with qPCR as well, with the designed LysV probe (5′–3′ TCGGACAACTGAGATAGCGG). The reference gene *GAPDH* was targeted with the following primers and probe (5′–3′): *GAPDH* forward ACTTGCCCTTCAAACGACAC, *GAPDH* reverse TCCACCACGTTCAAGGGATG, *GAPDH* probe ACGGTTTTGGACGTATTGGAC. The probes were modified with 5′ terminal reporter dyes (FAM for the viral target, Cy5 for *GAPDH*) and 3′ terminal quenchers (BHQ-1 for the virus, BHQ-2 for *GAPDH*). All reactions were carried out in triplicate using 10 µL reaction volumes with 1 µL template cDNA on an ABI 7500 Fast Real Time PCR system (Applied Biosystems, Foster City, CA, USA). Viral expression was measured relative to the reference gene by creating standard curves (Appendix A) for both viral and *GAPDH* RNA (linear range 1–10^−3^ ×), calculating primer efficiencies with the formula $10^{\left(\;-\frac{1}{slope}\;\right)}-1$ and then applying a comparative C_q_ analysis [46,47,48] to the qPCR results in R version 3.2.2. [49]. After finding some discrepancies between the PCR and qPCR assays, the amplicons from wild population samples screened with both PCR and real–time qPCR were Sanger sequenced (GATC Biotech AG, European Custom Sequencing Centre, Köln, GER).

### 2.4. Assembly of Viral Genome

LysV was initially detected from three transcripts that were highly differentially expressed between the experimentally-evolved populations of *L. fabarum*, collected as adults at generation 11 of the experiment by Dennis et al. [35]. We assembled all possible viral sequences to obtain the complete nucleotide sequence of the viral polyprotein. In total, we used 110 RNA-seq libraries, constructed from whole-adult, female wasps (National Center for Biotechnology Information, NCBI, Sequence Read Archive, SRA, PRJNA290156, Accessions SAMN03840107- SAMN03840049, [35]) and 4–5 day old larvae (NCBI PRJNA290156, SRA Accessions SAMN10024115- SAMN10024165 [50]). These were *de novo* assembled with Trinity v2014.07.17 [51]. Within this assembly, we identified 111 contigs which matched to the previously assembled viral transcript (c32773, [35]) by blastn (megablast, e-value < 1 × 10^−3^). To identify sequence variants within each individual library, we performed a second round of *de novo* assembly, this time using these 111 sequences as a reference for separate assemblies of the viral polyprotein from each of the 110 transcriptomic libraries, using the default genome-guided options in Trinity. We filtered the results of these independent assemblies to retain only the longest sequences (>5000 bp). Conserved protein domains were identified with a blastp search against the SwissProt database [52].

### 2.5. Phylogenetic Placement of LysV

The relationship of LysV relative to other known viruses was examined using a phylogeny based on the RdRP region, extracted from the two distinct viral polyproteins that were whole-genome assembled (see results). The highly conserved domains of the RdRP region have previously been used to construct informative phylogenies of RNA viruses [3,10,53,54,55,56,57,58,59,60]. A variety of taxa belonging to the order *Picornavirales* [3,61], including its closest blast match, the DcPV, were chosen, and the amino acid alignment we used was kindly provided by N. Dheilly [3] (Appendix A). Viral transcripts were translated to the existing amino acid alignment in Geneious version 6.1.8. [62].

An appropriate model of amino acid substitution was determined using ProtTest 2.4 [63] and subsequently implemented in phylogenetic analyses using Bayesian inference with Markov Chain Monte Carlo (Bayesian MCMC) and maximum likelihood. Bayesian inference of phylogeny was performed in MrBayes [64,65], run for 3,000,000 generations, with trees sampled every 100 generations, a burn-in of 25%, and repeated five times. The second highest scoring amino acid model (VT) based on ProtTest was selected, as the highest scoring model (HIV-W) was not available in MrBayes. Maximum likelihood phylogeny construction was performed with RAxML [66] with bootstrapping and implementing the HIV–W model with the PROTGAMMAI substitution model and a random number seed. The analysis was run for 1000 rapid bootstrap searches and repeated five times.

### 2.6. Transmission Assay

In addition to viral presence across wasp lineages, it is also important to determine the mode of viral transmission (horizontal or vertical). To test this, a transmission assay was set up with infected and uninfected wasp lines competing for the same aphid hosts, followed by rearing of their offspring (Figure 1). This enabled the detection of horizontal transmission events (infected offspring from uninfected mothers) as well as vertical transmission (infected offspring from infected mothers). Based on our screenings of lab–maintained wasp populations, the experimentally evolved line H76 was expected to be the most consistently infected population, and was chosen as the source of infected wasps. An independently reared asexual line that we found to be uninfected was used as the source of uninfected wasps (IL06-242). The two lines were distinguishable based on heritable morphological differences of forewings and femur [67]. After each generation, removed wasps were screened for viral RNA using PCR. Throughout the experiments, wasp and aphid lines were kept on single *Vicia faba* plants (pot volume of 0.07 l) under the same climatic conditions mentioned previously. All wasp lines in the transmission assay setup were reared on *H. defensa* free aphids of the genotype A08–28.

The transmission assay was run over three generations (Figure 1). Generation 1 (parental generation) was set up to confirm that individual mothers were infected or uninfected, so that they could be used to found the second generation. For this, single female wasps were caged on each plant with one week old host aphids, left to infect for 24 h (15 replicates of single female wasps of the asexual line IL06–242 and 30 replicates of the H76 line). The second generation was used in the assays that combined infected and uninfected sexual and asexual wasp lines together on the same hosts. For this second generation, wasps from the sexual and asexual lines were sampled at the same time each day over a four day period and the H76 line sorted based on gender (Figure 1).

Three different types of wasp combinations were set up in the second generation. Assay type 1 contained female–female pairs (seven replicates) with only one infected H76 and one uninfected asexual female that were removed from the aphid hosts after 24 h. These female-female pairs were set up to trace the viral infection by direct screening through all three generations with knowledge of the specific mother-progeny combination. Type 2 assays contained male–female pairs (four replicates) with one infected male (H76) and uninfected female (asexual), and parental wasps were removed from the hosts after 24 h. The male–female pools completely excluded two possible modes of transmission (superparasitism and mutual stinging), since males cannot inject eggs into aphid hosts. However, here it was possible that there is transmission by mating. The asexual females, although capable of reproducing asexually, may be forced to mate in combination with sexual males, possibly leading to viral transmission through the infected male. Assay type 3 contained larger pools of wasps (six replicates), with ≥4 of both infected (H76) and uninfected (asexual) individuals, and included some pools with only H76 females and some with H76 females and males (Figure 1). In the type 3 assays, the parental wasps were not removed, meaning they had >24 h to infect hosts. These larger pools were set up with the aim to include as many viral transmission modes as possible. If the virus can be transferred to the uninfected asexual lines by superparasitism [68,69] or mutual stinging the chance of this occurring is greater if there is a higher density of wasps on the same number of aphid hosts. Each of the three setups harbored the possibility of horizontal transmission through airborne particles or direct contact of the wasps. Upon emergence of the last generation the wasps were sampled at the same time every day over a period of five days (wasps between 0–24 h old) to reduce variation due to daily fluctuations and age differences.

### 2.7. Statistical Analyses

We tested for differences in viral infection frequencies among wasp lines based on PCR results using Fisher’s Exact tests for count data. Differences in viral load based on qPCR (whole wasp samples and ovary dissections tested separately) between wasps sampled in April and October 2016 were tested for their statistical significance (*p* < 0.05) with a mixed linear model (Appendix A). The calculated mean amount of viral RNA per *GAPDH* (log transformed) for individual wasps served as the response variable. Wasp lines were considered separately and only lines with samples available for both sampled time points were compared. Unbalanced data was accounted for by using the Satterthwaite approximation for degrees of freedom. *p*-values of the fixed effect (generation) were calculated using the F–test and those of the random effect (sample within generation) using the likelihood ratio chi-square test. Samples of the transmission assay were tested for statistically significant differences in viral load also using a mixed model (Appendix A) with the same response variable. Horizontal transmission in generation three individuals, vertical transmission of H76 individuals across all three generations and vertical transmission across only the first two generations were analyzed separately. Pools were considered individually, as their setup differed. Statistical analysis was performed in R version 3.2.2. using the lme4 [70] and lmerTest [71] packages.

## 3. Results

### 3.1. Lab Populations

Based on PCR screenings, the infection percentages of wasp lines sampled from the lab in April 2016 varied widely. Of the five experimental evolution lines, three (H76, H76 reared on H– and H402 on H-) showed viral presence (91.8%, 50% and 25% infection percentage, respectively) while no infected individuals were detected in the other two lines (Table 1). Two of the five tested asexual lines revealed a presence of the virus (IL07-64 and IL09-554 with 33.3% infection percentage each). The mixed, sexual population displayed a high infection percentage of 75%. *p*-values for differences in viral infection frequencies between lines are shown in Appendix A. Comparing these infection results to a later time point (October 2016, 16 wasp generations) revealed a decline in the percentage of infected individuals in various populations (Table 1). However, the only significant declines we observed were in one experimental-evolution population (H76) and in the two asexual lines previously found to carry the virus. The large sexual population was the only population that showed an increasing trend in percentage of infected individuals over the same time period. Importantly, we did not detect LysV in the host aphids, nor did we find the viral sequence in blastx searches of transcriptomic data generated from wasp-free aphids [50].

Viral screening with qPCR revealed differences from the standard PCR assay. While standard PCR detected virus in three lines, an additional line had detectable virus using qPCR (H402, Figure 2). All individuals that tested virus positive with only qPCR had a much lower viral load than those testing positive with both methods, illustrating the higher sensitivity of the qPCR assay. The virus was detected at both tested time points with qPCR. Viral load varied significantly between individual wasp samples (for tested populations present in April and October 2016, *p*-values ≤ 0.008), but not between time points (*p*-values ≥ 0.1) (*p*-values in Appendix A).

With this more sensitive qPCR method, viral RNA was also found in three asexual wasp samples, previously tested negative with PCR. The virus was detected at both time points in line IL09–554 (differences between time points not significant, *p*-value = 0.259) and at one time point for line IL07–64. No viral presence was ever observed in line IL06–242, used as the uninfected line in the transmission assays. Further, dissected wasp samples (ovaries and bodies) also demonstrated this higher sensitivity of the qPCR assay. Out of 18 dissected samples chosen for qPCR screening, eight additional samples tested positive for the virus, including one sample from the H– line, where no viral presence was previously found.

### 3.2. Wild Populations

We detected the virus in wild populations using both standard PCR and qPCR screenings. Of 368 wild collected wasps tested, seven individuals were positive for the virus (Table 2). One positive sample came from Orbe, Switzerland, and was collected in 2009. All remaining positive individuals were collected in 2012 and came from the preserved material of those samples that were also used to establish the laboratory stocks for the experimental evolution study of Dennis et al. [35]. Two positive 2012 samples were obtained from material that was preserved after the wasp lines had been established and reared in the laboratory. The origin of the virus in the lab populations was, therefore, likely to be from these samples collected in 2012. The four sites yielding virus-positive samples are all in close proximity to each other and limited to a region in Western Switzerland (Figure 3). The virus thus does not seem to be present in very high abundance in the sampled regions and is most likely represented in higher numbers in the lab due to a bottleneck when establishing lab populations. However, the virus was still detected in four independent populations in the wild, demonstrating that it has potential to impact wild wasps and could be prevalent in some areas.

Testing the seven PCR positive wild population samples with qPCR revealed unexpected patterns. Five of these previously positive samples yielded a negative result with qPCR (samples from Fribourg and Nyon, Figure 4). Only the samples from Orbe and Renens were also positive with qPCR, with the viral load of the sample from Orbe being much lower than that of the Renens sample, which corresponded well with the weaker band obtained from standard PCR (Figure 4). This discrepancy between standard PCR and qPCR results was suggestive of some sequence variation compromising our qPCR assay.

To address this, we Sanger sequenced the PCR amplicons shown in Figure 4 that correspond to the infected samples shown in Table 2. Indeed, in the binding region of the designed qPCR TaqMan probe, three different haplotypes were identified: the viral sequences from the Orbe and Renens sample were identical and matched the designed LysV probe (hereafter referred to as haplotype A), all the individuals from the Nyon samples were infected with a different haplotype (B) and the Fribourg sample carried a third haplotype (C) (Figure 5). Additional point mutations occurred outside the probe’s binding region. Fortunately, there was no sequence variation in the binding sites of primers LysV forward and reverse, explaining positive PCR detection of all viral haplotypes, while the LysV qPCR probe is haplotype specific. We were able to construct the full viral genome of haplotypes A and B (see below) but haplotype C was not present in high enough abundance in our data.

### 3.3. Description of the Viral Genome

We assembled the complete viral polyprotein using Illumina reads sequenced from the experimentally evolved populations of *L. fabarum*. Across the 110 RNA-seq libraries, separate assemblies of viral reads yielded 2862 sequences. However, many of these were identical to one another and were short and incomplete (>2500 below 1000 bp). To examine only the nearly complete polyproteins, we subsetted the 48 longest sequences (>5000 bp). Within these, there were two distinct LysV polyproteins, which we have labelled “A” and “B” to match with the Sanger sequencing described above (National Center for Biotechnology Information, NCBI, Genbank, Accessions MK682509–MK682510). Among the 48 long sequences, LysV A and LysV B were equally represented (26 and 22 sequences, respectively). These two sequences differ by ca. 650 nucleotide variants (113 non-synonymous, Appendix A) but there is low diversity within each type (8–9 variable positions, 2–3 non-synonymous), and we cannot exclude that these are sequencing errors. The GC content was similar between the two types (LysV A: 35.1%, LysV B: 35.3%). We did not find evidence of any individual population containing more than one viral type, however, it is very likely that we have excluded low-abundance viral types by analyzing only the longest sequences, which presumably represent the most abundant polyproteins. We only found LysV A in wasp lines reared in aphids from the H402 treatment. Both the H76 and H– treatments contained independent replicate wasp populations with either LysV A or LysV B. Comparisons at the protein level with the Swiss-prot database identified five conserved proteins in the viral polyprotein: three putative capsid proteins, an RNA helicase, and the full RdRp region (Figure 6). Similar to other Iflaviridae, structural proteins appear towards the 5′ end of the polyprotein, while the RdRp is very near to the N-terminus [9,72].

### 3.4. Phylogenetic Placement of LysV

Both Bayesian inference and maximum likelihood analyses of the alignments of the RdRP region produced identical tree topologies and classified the RNA virus (haplotypes LysV A and LysV B) as belonging to the Iflaviridae family in the order of the Picornavirales (Figure 7). Its closest relative was the *Venturia canescens* picorna-like virus (VcPLV) [29], followed by the *Dinocampus coccinellae* paralysis virus (DcPV) [3] and the *Nasonia vitripennis* virus isolate 1 (NvitV–1) [30], all part of the Iflaviridae family.

### 3.5. Transmission of LysV

We observed evidence of both horizontal and vertical virus transmission, but horizontal transmission rates were much lower than vertical. Since it was most consistently infected, we were able to test and detect vertical transmission in all three assay types using the H76 line (Figure 1). Based on standard PCR screenings, the overall infection percentage of the H76 wasps varied between 11.8% and 50.7% in the three generations across all 17 pools tested (Table 3). This showed that vertical transmission was imperfect, as not all offspring of infected mothers were positive for the virus in generation 3. In the female-female pairs (assay type 1, *n* = 20), 50–100% of third-generation H76 offspring were infected.

We chose the previously uninfected IL06-242 line (Table 3) to test for horizontal transmission. Based on our design, we could only detect potential horizontal transmission in generation 3. Across all assay types (seven female-female pairs, four male-female pairs, six large pools) standard PCR screening (Table 3) identified two newly-infected asexual individuals, one each in two pools of assay type 1 (female–female pairs). Subsequent analysis of these pools with qPCR proved to be more sensitive for viral detection than standard PCR screenings, and seven additional third-generation asexual wasps tested positive with qPCR. These wasps were found in the same two female-female pairs, making a total of nine instances of newly established horizontal infection in two separate test pairs (out of seven total tested female-female pairs). Lastly, in the large pool assays (Type 3 assay), we also detected horizontal transmission in a single, additional asexual wasp that was infected (out of six pools tested).

As with vertical transmission, the horizontal transmission we detected was also imperfect. In each female-female pair, four of the five asexual wasps in generation 3 were virus positive (based on qPCR), and in the large pool assay only one individual was detected (Figure 8). Importantly, no transmission was detected in male–female pairs, suggesting that there is no virus transmission through mating.

## 4. Discussion

### 4.1. A New RNA Virus

Our analyses of RNA obtained from the aphid parasitoid *L. fabarum* revealed a previously undescribed species of RNA virus, which we have named *Lysiphlebus fabarum* virus (LysV). Prior to this study, blast results showed a close similarity of the LysV polyprotein to the polyprotein of DcPV [35], an RNA virus with no DNA stage [3]. Phylogenetic analyses confirmed this and classified LysV as belonging to the Iflaviridae, an invertebrate infecting RNA virus family in the order Picornavirales [61,73]. The classification of LysV in this family aligns with the main characteristics describing this group, namely that the viral genomes consist of positive single strand RNA molecules [73], and that the viral genomes encode RdRP for replication of their genetic material, [73]. All of the Iflaviridae known to this day have been isolated from arthropod species and have a restricted host range [73]. Finally, there are no matches to this polyprotein in the sequenced genome of the wasp parasitoid *L. fabarum* [74], indicating that LysV does not reverse transcribe, nor does it integrate itself into the genome of its wasp host.

### 4.2. Viral Haplotype Variation in Wild Populations

Three different virus haplotypes were identified in field-collected *L. fabarum*, of which two were also present in the laboratory populations we studied. This diversity could be due to the rapid evolution rate that has been documented in viruses [75,76]. The different viral haplotypes found also point towards a long-term presence of the virus in the wild. During this time, different haplotypes could have diverged and established themselves across wasp populations. It is of note that there is relatively low diversity within each haplotype, suggesting selection could maintain the separate haplotypes. In the transcriptome of Dennis et al. [35] only two viral haplotypes (type A and B) were detected. For types A and B the abundant transcriptomic data allow us to reconstruct the entire polyprotein, and compare nucleotide variation across it. With the haplotypes we have detected, there are three layers of viral variation: nucleotide variation among haplotypes, individual variation in viral load between wasps, and population-level variation in infection percentage. All wasps testing positive for the virus belonged to the species *L. fabarum*, but it is very well possible that other aphid parasitoids carry LysV since other viruses, such as the *Diadromus pulchellus* ascovirus (DpAV), have been detected in more than one species belonging to the Ichneumonidae [77].

### 4.3. Symptoms

No obvious positive or negative effects have been observed in infected populations/individuals compared to non-infected ones. This points towards a persistent non-symptomatic infection of the virus, as has been reported for other viruses infecting parasitoids [29,30,78]. Further targeted experiments to detect viral symptoms would be needed to fully establish whether the virus causes effects that are not obvious at first glance, both harmful or favorable for the wasp. RNA viruses of other hymenopteran parasitoids have been found to be beneficial for the wasp during host infection by making infected hemocytes of the host unable to encapsulate the parasitoid eggs [79]. In the case of DcPV, shown to be closely related to the virus described here, the virus helps its carrier by usurping the natural host behavior, which is directly correlated to the presence of viral RNA [3]. A different example shows the severity of RNA viruses as pathogens in hymenopteran parasitoids: in the endoparasitic wasp *Pteromalus puparum*, an RNA virus leads to the deterioration of the venom apparatus [80]. These examples underline the fact that virus–parasitoid interactions can span all aspects of symbiotic relationships: parasitism, commensalism and mutualism. As the three most closely related viruses (VcPLV, DcPV, and NvitV–1) show commensal to mutualist relationships [3,29,30], it is possible that the virus in *L. fabarum* will also fall into one of these categories. It is also possible that further examination of virus-infected individuals will reveal that it has an influence on parasitoid success or behavior. The higher presence of LysV in experimentally evolved lineages with different infective abilities suggests that this virus could be advantageous in some settings [35]. However, this could also be the result of the bottlenecks experienced during this experiment. Future work should explicitly test the relationship between parasitoid infectivity and viral load.

### 4.4. Viral Transmission

The transmission assays revealed that LysV can be both vertically transferred between mothers and offspring of the same lines as well as horizontally transmitted to individuals of different genotypes. Vertical transmission was more frequent than horizontal transmission, but both modes were imperfect. With mounting evidence of viruses infecting organisms with short generation times, such as arthropods, the importance of vertical transmission has been emphasized as a means of ensuring viral passage [3,11,77,81,82,83]. Yet, horizontal transmission is also important in this system, particularly as the primary means of transmitting to genotypes that only reproduce asexually.

Distinguishing clearly between the different forms of transmission (horizontal and vertical) can be difficult in systems such as these parasitoids because of their close proximity (in or on) their aphid hosts, and the possibility of multiple parasitoids attempting to infect the same host (superparasitism). Vertical transmission was observed in experimental setups containing only infected females, suggesting that the most likely form of vertical transmission is maternally through the eggs, but we could not distinguish this from viral deposition in the host followed by re-infection of the simultaneously injected egg. Maternal transmission through eggs has been observed in combination with horizontal transmission in a virus infecting the parasitoid wasp *L. boulardi* [69], blurring the distinction between the two transmission possibilities. If superparasitism occurs in a host and one egg is infected and the other uninfected, the virus can be transmitted horizontally to the uninfected egg [68,69] while still appearing to be vertically transmitted by the mother. This route of infection is plausible in *L. fabarum*, as we have observed superparasitism in the lab [35]. Visualizing viral particles in different tissues through transmission electron microscopy could help distinguish these possibilities and determine where exactly the virus is located in hosts. Further plausible transmission possibilities include horizontal transmission through mutual stinging among wasps, often visible when wasps are crowded in a smaller area. The transmission pathways that seem the most unlikely are horizontally transmitting the virus through direct contact of the wasps, through air–borne particles or via infected males during mating. Males never succeeded in passing on the virus to uninfected females in our experiment, and many laboratory lines were uninfected despite being reared for many generations in close physical proximity of virus-infected lines.

The newly-infected asexual individuals obtained from the transmission assay all displayed a rather low viral load (only two of nine individuals showed a high enough amount of viral transcription to be detected by standard PCR). This hints at a reduced viral load in newly-infected individuals, and the future outcome of these new infections needs to be investigated more thoroughly. High viral loads do not seem to be a requirement for transmission, however, as high fluctuations in viral loads across all generations of the transmission assay were observed: parental generations with very low viral loads could produce highly infected offspring and vice versa. Based on this, we conclude that any level of viral presence can lead to vertical transmission, but that not all offspring of infected parents carry the virus. If the high rates of vertical transmission failures we observed are representative of the natural situation, it would seem that they have to be compensated by relatively high rates of horizontal transmission or by some selective advantage of infected parasitoids for the virus to persist in host populations.

## Figures and Tables

**Figure 1 viruses-12-00059-f001:**
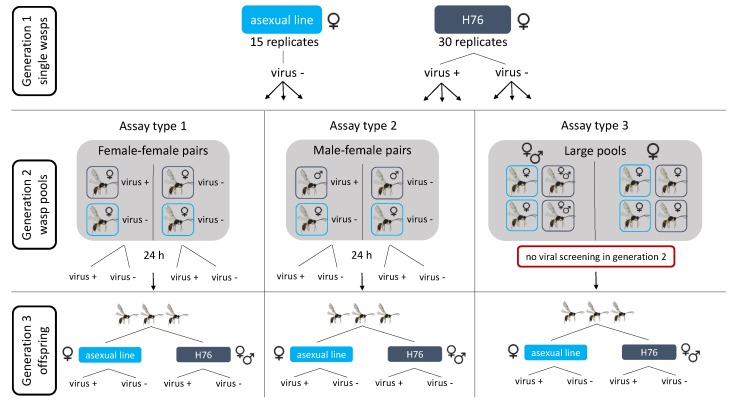
Experimental setup of wasp pools for the transmission assay, to test possible modes of transmission (horizontal and vertical) with the wasp lines reared for three generations. Generation 1 consisted of single female wasps reared on aphid hosts. The uninfected asexual line was combined with infected and uninfected individuals of the H76 line to create either female–female, male–female pairs, or large pools for generation 2. Wasps in pairs were put on aphid hosts for 24 h before being removed and screened for the virus. Wasps in large pools were not removed nor screened for the virus at generation 2. Generation 3 consisted of sampling the offspring of the pooled lines, sorting them based on line and subsequently testing for the presence of the virus.

**Figure 2 viruses-12-00059-f002:**
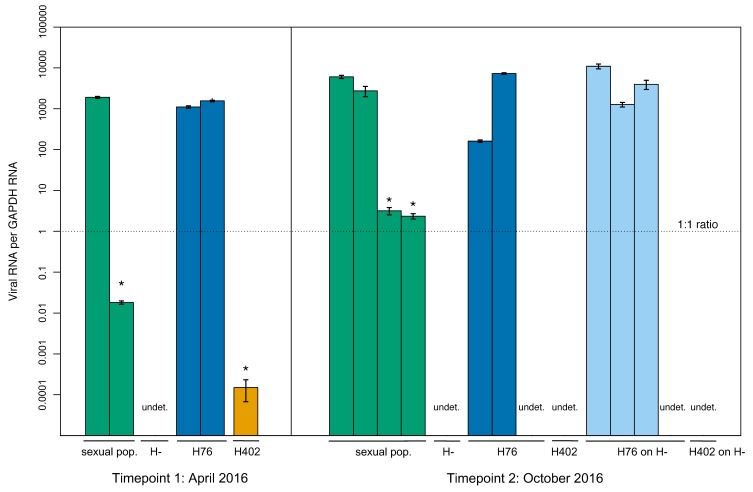
Lab population samples tested with qPCR. Bars show the viral loads of individual wasps per *GAPDH* ± standard error on a log scale. Virus was detected in all lines except H– and H402 on H– (only one undetected sample each shown). The amount of viral RNA in samples varies highly. Asterisks show samples that were negative based on PCR but show viral presence based on qPCR. The dashed horizontal line shows where the number of viral and *GAPDH* transcripts are equal and samples with no viral detection are labelled with “undet.”. Colors depict the different lines as follows: sexual population (green), H76 (blue), H402 (orange), and H76 on H– (light blue).

**Figure 3 viruses-12-00059-f003:**
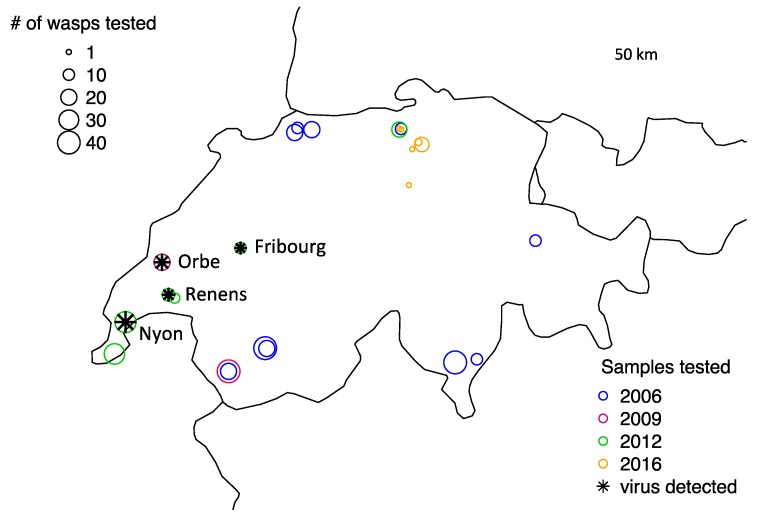
Swiss sampling sites where wasps were collected and tested for LysV (five additional sampling countries not shown). Viral presence was observed at four sites (shown by the asterisks), all in Western Switzerland. The samples were collected during different sampling events, indicated by the colors: 2006 (blue), 2009 (magenta), 2012 (green), and 2016 (orange). The area of the points correlates with the number of wasps extracted and tested for the virus.

**Figure 4 viruses-12-00059-f004:**
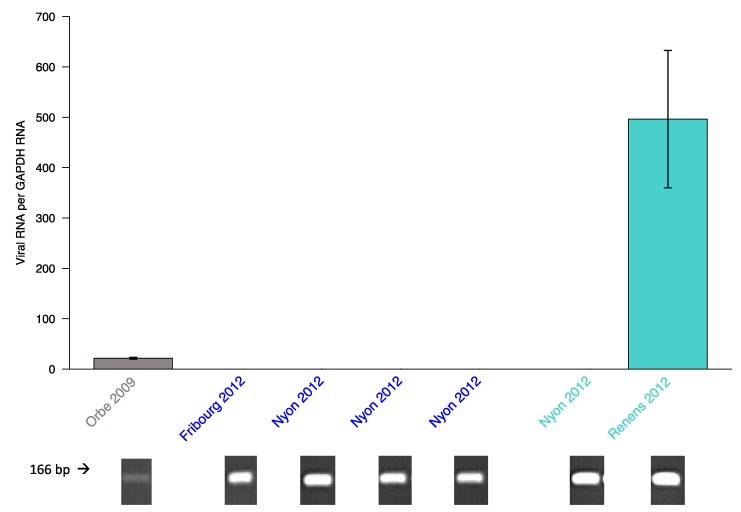
Discordance between qPCR and PCR assays. qPCR results of the wild individuals where PCR detected viral presence. Five of the previously positive samples showed no detection of the virus. Bars show the mean relative viral load per sample ± standard error. The colors describe the populations as follows: wild populations (gray), founding populations of evolution lines before maintenance in the lab (blue) and founding populations of evolution lines after maintenance in the lab for 30 generations (turquoise).

**Figure 5 viruses-12-00059-f005:**
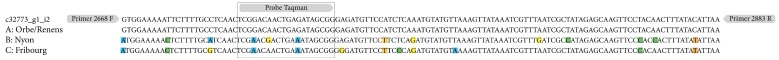
Sequencing results of the different wild populations. Three different haplotypes were identified, with point mutations being present in the binding region of the LysV probe. Samples from Orbe and Renens carry the same haplotype (**A**), matching the designed probe, while the samples from Nyon (**B**) and Fribourg (**C**) each have a different haplotype. The original transcript the primers were designed from can be seen at the top (c32773_g1_i2).

**Figure 6 viruses-12-00059-f006:**
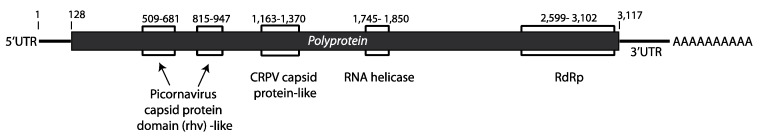
Diagram of putative organization of the LysV genome. Predicted proteins are marked with black boxes. Numbers above indicate nucleotide positions.

**Figure 7 viruses-12-00059-f007:**
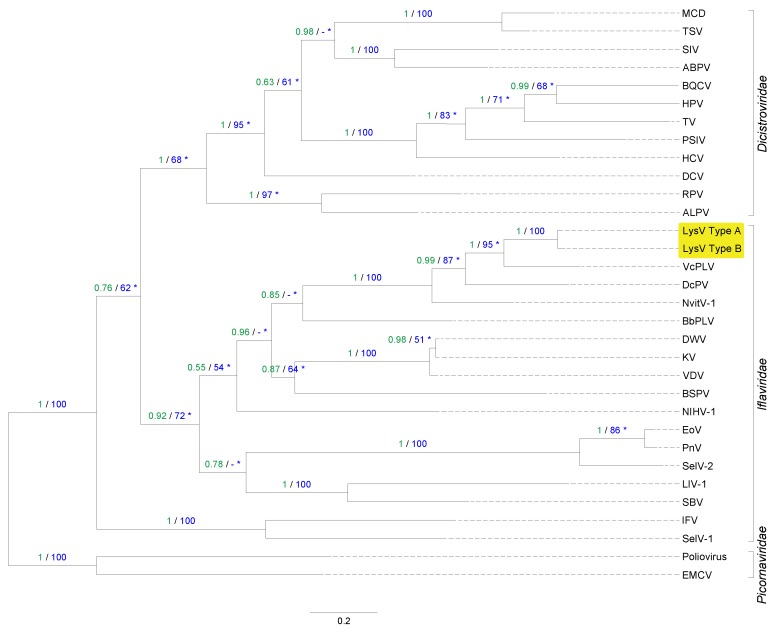
Phylogenetic tree showing that both haplotypes of the detected virus (LysV highlighted in yellow) belong to the *Iflaviridae* family (order *Picornavirales*). The tree was constructed from amino acid alignments of the RdRP gene and is shown with Bayesian posterior probabilities (green) and maximum likelihood bootstrap support values (blue). Values of at least 0.5% and 50%, respectively, are indicated. Asterisks indicate non-identical bootstrap support values between maximum likelihood runs. If this is the case, the single highest value is shown (replicate values did not differ by more than 6%). The scale bar is proportional to the number of amino acid substitutions per site.

**Figure 8 viruses-12-00059-f008:**
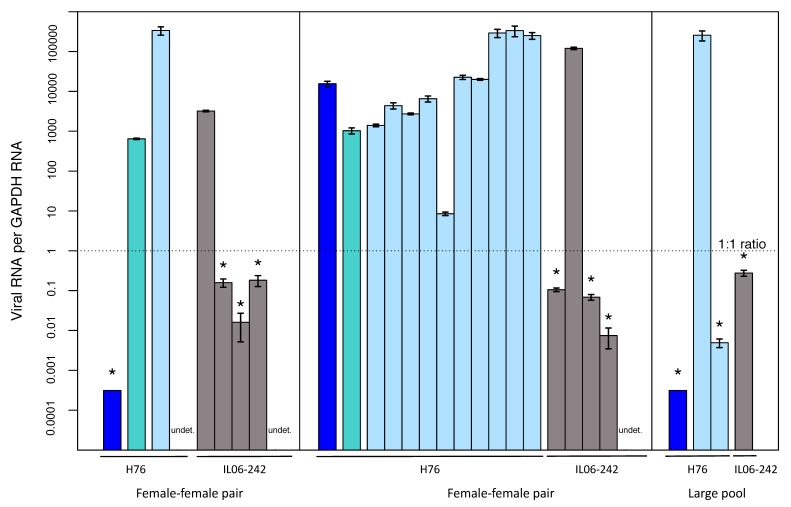
Transmission assay results: pools with observed horizontal transmission tested with qPCR. Bars represent the relative viral load per *GAPDH* ± standard error of an individual wasp and are shown on a log scale. Nine asexual individuals were newly infected across the three pools. Samples negative for PCR but positive for qPCR are labelled with asterisks and the dashed line displays where viral and *GAPDH* transcription are equal. Generations and lines are labelled by colors as follows: H76 generation 1 (blue), H76 generation 2 (turquoise), H76 generation 3 (light blue) and IL06–242 generation 3 (gray). Only H76 individuals are shown for generations 1 and 2, as all asexual individuals were negative for the virus in these generations. All emerged individuals, positive or negative for the virus, are shown for both lines in generation 3. No generation 2 individuals are shown for the large pool as wasps were not removed and hence not tested for this generation.

**Table 1 viruses-12-00059-t001:** Comparison of the infection rates of all tested lab populations between two time points (April and October 2016) by PCR screening.

Population		April 2016	October 2016	Fisher’s Exact Test
Experimental evolution populations	H–	0% (*n* = 10)	0% (*n* = 18)	*p* = 1
	H76	91.8% (*n* = 98)	19.1% (*n* = 21)	*p* < 0.001
	H402	0% (*n* = 78)	0% (*n* = 17)	*p* = 1
	H76 on H–	50% *(n* = 8)	17.6% (*n* = 17)	*p* = 0.156
	H402 on H–	25% (*n* = 4)	0% (*n* = 19)	*p* = 0.174
Asexual populations	IL06-242	0% (*n* = 15)	0% (*n* = 20)	*p* = 1
	IL06–680	0% (*n* = 15)	0% (*n* = 20)	*p* = 1
	IL07–64	33.3% (*n* = 15)	0% (*n* = 20)	*p* = 0.009
	IL09–402	0% (*n* = 15)	0% (*n* = 20)	*p* = 1
	IL09–554	33.3% (*n* = 15)	0% (*n* = 20)	*p* = 0.009
Sexual population	sexual mixed population	75% (*n* = 20)	92.9% (*n* = 42)	*p* = 0.098

**Table 2 viruses-12-00059-t002:** Number of virus positive samples of wild populations.

Population		Number of Wasps Tested (PCR)	Number of Samples Infected (PCR)
Wild populations	2006	250	0
	2009	40	1
	2012	12	0
	2012 founding populations before lab	17	4
	2012 founding populations after lab	36	2 (samples of 4 wasps each)
	2016	13	0

**Table 3 viruses-12-00059-t003:** Infection percentages of wasp lines used for the setup of the transmission assay. Only H76 wasps were already virally infected at generation 1.

Population	Generation	# of Wasps Tested (PCR)	# of Wasps Infected (PCR)	Infection Percentage	Type of Transmission
H76 (sexual)	**1**	28	4	14.3%	Horizontal
	**2**	71	36	50.7%	Vertical and horizontal
	**3**	93	11	11.8%	Vertical and horizontal
IL06-242 (asexual)	**1**	13	0	0%	NA
	**2**	15	0	0%	NA
	**3**	82	2	0.02%	Horizontal

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
