# Peer review of "A Novel RNA Virus in the Parasitoid Wasp Lysiphlebus fabarum: Genomic Structure, Prevalence, and Transmission"

_viruses, 2020, doi:10.3390/v12010059_

Round 1
Reviewer 1 Report
In this study, the LysV was found based on the transcriptome sequences of wasps from an experimental evolution study. The authors screened the LysV prevalence in the field and laboratory lines and also the vertical and horizontal transmission of LysV were demonstrated experimentally. However, based on the data, there are some major concerns need to be clarified for this study:
1) It seems the virus only have sequence-based data, however, it is very important to see the really viral particle in the filed or laboratory lines line sample by either negative staining with TEM to confirmed the real viral particles
2) For the vertical and horizontal transmission of LysV, is there any tissue section, which was observed by TEM? It’s very important for demonstrating the transmission of LysV
3) For the viral genomic sequence, how to make sure the complete genomic sequence of the RNA virus? Does the 5’ and 3’ end RACE perform?
4) The protein region should be predicted in more detail. Is there any protease region in the genome? What is the identity of the non-structural and structural proteins to other close-related viruses?
Reviewer 2 Report
Editors,
The manuscript entitled, “A novel RNA virus in the parasitoid wasp Lysiphlebus fabarum: genomic structure, prevalence and transmission ” by Luhi et. al describes a recently discovered RNA virus in parasitoid wasps. This manuscript contains data from laboratory-based transmission studies – and thus goes beyond a simple genome description.
Points to clarify or address before publication include:
Lines 27-28 Abstract – since symptoms may be difficult to discern in parasitoid wasps, it seems the last line of speculation (i.e., lines 27-28) should be removed from the abstract – though they would be fine to have in the discussion section. Figure 2 – the orientation was wrong in the draft I review and some of the text too small and narrow to read. Table 2 – with so few positive PCR tests, all should be sequenced verified because the primer set may pick up other viral or host products Figure 4 – is this virus only important in the lab? It doesn’t seem like is was detected much outside of laboratory samples. Was negative strand specific PCR done to confirm virus replication? This would be good to do on some samples. Make sure that the text near the qPCR description of haplotypes is based on having mostly complete genomes of at least two haplotypes based on the RNASeq data – and include NCBI reference numbers for each of the haplotypes described in this study.When reading the qPCR section, I was concerned that sequencing only 166 bp of sequence – with differences would not constitute a haplotype, particularly since RNA viruses with error prone polymerases exist in quasi-species. Was there LysV in the aphids too? The methods should include the linear range of the qPCR assay for LysV and linear equation for primer efficiency tests (i.e., dilutions of virus containing stock) as described in Ginzinger et al 2002 - https://doi.org/10.1016/S0301-472X(02)00806-8 Or other comparable reference. For qPCR assays – were standard curves done for both viral RNA and GAPDH RNA, since that would be the only way to get the viral RNA per GAPDH RNA. Please elaborate onhow you go from Ct values to copy numbers without a standard curve and/or include information on standard curves from which you can calculate relative abundance, and then compare “starting quantities” of RNA at the copy number level.Were the efficiency tests for GAPDH and LysV exactly the same, so Ct values were directly compared? Please inclue linear range of qPCR assays, and linear equations in the methods. In addition to assembling the genome from RNASeq data using Trinity, were large segments (~ 1,000 nt or more) PCR- amplified and Sanger sequenced to confirm assembly? If not, some of the sequence / assembly (or larger genome segments) should be PCR-confirmed and sequenced.
Minor points to clarify or address before publication include:
Refs 9-11 – Could include recent Annual Review in Entomology on honey bee viruses by Grozinger and Flenniken. https://doi.org/10.1146/annurev-ento-011118-111942 Line 53 – H. defense should be italicized
